# CAN CNNS BE MORE ROBUST THAN TRANSFORMERS?

**Zeyu Wang**[1] **Yutong Bai**[2] **Yuyin Zhou**[1] **Cihang Xie**[1]
[1]UC Santa Cruz [2]Johns Hopkins University

## ABSTRACT

The recent success of Vision Transformers is shaking the long dominance of Convolutional Neural Networks (CNNs) in image recognition for a decade. Specifically, in terms of robustness on out-of-distribution samples, recent research finds that Transformers are inherently more robust than CNNs, regardless of different training setups. Moreover, it is believed that such superiority of Transformers should largely be credited to their *self-attention-like architectures per se*. In this paper, we question that belief by closely examining the design of Transformers. Our findings lead to three highly effective architecture designs for boosting robustness, yet simple enough to be implemented in several lines of code, namely a) patchifying input images, b) enlarging kernel size, and c) reducing activation layers and normalization layers. Bringing these components together, we are able to build pure CNN architectures without any attention-like operations that are as robust as, or even more robust than, Transformers. We hope this work can help the community better understand the design of robust neural architectures. The code is publicly available at https://github.com/UCSC-VLAA/RobustCNN.

## 1 INTRODUCTION

The success of deep learning in computer vision is largely driven by Convolutional Neural Networks (CNNs). Starting from the milestone work AlexNet (Krizhevsky et al., 2012), CNNs keep pushing the frontier of computer vision (Simonyan & Zisserman, 2015; He et al., 2016; Tan & Le, 2019). Interestingly, the recently emerged Vision Transformer (ViT) (Dosovitskiy et al., 2020) challenges the leading position of CNNs. ViT offers a completely different roadmap—by applying the pure self-attention-based architecture to sequences of image patches, ViTs are able to attain competitive performance on a wide range of visual benchmarks compared to CNNs.

The recent studies on out-of-distribution robustness (Bai et al., 2021; Zhang et al., 2022; Paul & Chen, 2022) further heat up the debate between CNNs and Transformers. Unlike the standard visual benchmarks where both models are closely matched, Transformers are much more robust than CNNs when testing out of the box. Moreover, Bai et al. (2021) argue that, rather than being benefited by the advanced training recipe provided in (Touvron et al., 2021a), such strong out-of-distribution robustness comes inherently with the Transformer's self-attention-like architecture. For example, simply "upgrading" a pure CNN to a hybrid architecture (*i.e.*, with both CNN blocks and Transformer blocks) can effectively improve out-of-distribution robustness (Bai et al., 2021).

Though it is generally believed that the architecture difference is the key factor that leads to the robustness gap between Transformers and CNNs, existing works fail to answer which *architectural elements* in Transformer should be attributed to such stronger robustness. The most relevant analysis is provided in (Bai et al., 2021; Shao et al., 2021)—both works point out that Transformer blocks, where self-attention operation is the pivot unit, are critical for robustness. Nonetheless, given a) the Transformer block itself is already a compound design, and b) Transformer also contains many other layers (*e.g.*, patch embedding layer), the relationship between robustness and Transformer's architectural elements remains confounding. In this work, we take a closer look at the architecture design of Transformers. More importantly, we aim to explore, with the help of the architectural elements from Transformers, *whether CNNs can be robust learners as well*.

Our diagnose delivers three key messages for improving out-of-distribution robustness, from the perspective of neural architecture design. Firstly, patchifying images into non-overlapped patches can

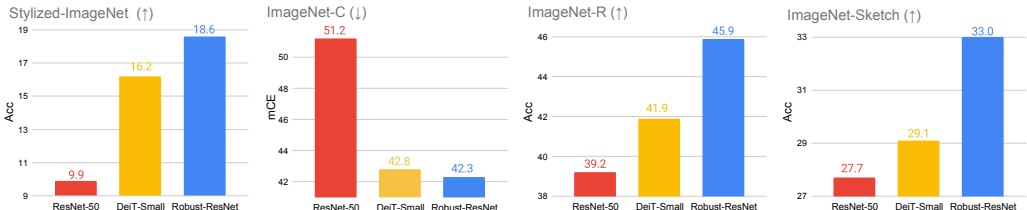

Figure 1: Comparison of out-of-distribution robustness among ResNet, DeiT, and our enhanced ResNet (dubbed *Robust-ResNet*). Though DeiT-S largely outperforms the vanilla ResNet, it performs worse than Robust-ResNet on these robustness benchmarks.

substantially contribute to out-of-distribution robustness; more interestingly, regarding the choice of patch size, we find the larger the better. Secondly, despite applying small convolutional kernels is a popular design recipe, we observe adopting a much larger convolutional kernel size (*e.g.*, from $3 \times 3$ to $7 \times 7$, or even to $11 \times 11$) is necessary for securing model robustness on out-of-distribution samples. Lastly, as inspired by the recent work (Liu et al., 2022), we note aggressively reducing the number of normalization layers and activation functions is beneficial for out-of-distribution robustness; meanwhile, as a byproduct, the training speed could be accelerated by up to ~23%, due to fewer normalization layers being used (Gitman & Ginsburg, 2017; Brock et al., 2021).

Our experiments verify that all these three architectural elements consistently and effectively improve out-of-distribution robustness on a set of CNN architectures. The largest improvement is reported by integrating all of them into CNNs' architecture design—as shown in Fig. 1, without applying any self-attention-like components, our enhanced ResNet (dubbed Robust-ResNet) is able to outperform a similar-scale Transformer, DeiT-S, by 2.4% on Stylized-ImageNet (16.2% *vs.* 18.6%), 0.5% on ImageNet-C (42.8% *vs.* 42.3%), 4.0% on ImageNet-R (41.9% *vs.* 45.9%) and 3.9% on ImageNet-Sketch (29.1% *vs.* 33.0%). We hope this work can help the community better understand the underlying principle of designing robust neural architectures.

## 2 RELATED WORKS

**Vision Transformers.** Transformers (Vaswani et al., 2017), which apply self-attention to enable global interactions between input elements, underpins the success of building foundation models in natural language processing (Devlin et al., 2019; Yang et al., 2019; Dai et al., 2019; Radford et al., 2018; 2019; Brown et al., 2020; Bommasani et al., 2021). Recently, Dosovitskiy *et al.* (Dosovitskiy et al., 2020) show Transformer attains competitive performance on the challenging ImageNet classification task. Later works keep pushing the potential of Transformer on a variety of visual tasks, in both supervised learning (Touvron et al., 2021a; Yuan et al., 2021; Liu et al., 2021; Wang et al., 2021; Yuan et al., 2021; Zhai et al., 2021; Touvron et al., 2021b; Xue et al., 2021) and self-supervised learning (Caron et al., 2021; Chen et al., 2021; Bao et al., 2022; Zhou et al., 2022; Xie et al., 2021; He et al., 2022), showing a seemly inevitable trend on replacing CNNs in computer vision.

**CNNs striking back.** There is a recent surge in works that aim at retaking the position of CNNs as the favored architecture. Wightman *et al.* (Wightman et al., 2021) demonstrate that, with the advanced training setup, the canonical ResNet-50 is able to boost its performance by 4% on ImageNet. In addition, prior works show modernizing CNNs' architecture design is essential—by either simply adopting Transformer's patchify layer (Trockman & Kolter, 2022) or aggressively bringing in every possible component from Transformer (Liu et al., 2022), CNNs are able to attain competitive performance w.r.t. Transformers across a range of visual benchmarks.

Our work is closely related to ConvNeXt (Liu et al., 2022), while shifting the study focus from standard accuracy to robustness. Moreover, rather than specifically offering a unique neural architecture as in (Liu et al., 2022), this paper aims to provide a set of useful architectural elements that allows CNNs to be able to match, or even outperform, Transformers when measuring robustness.

**Out-of-distribution robustness.** Dealing with data from shifted distributions is a commonly encountered problem when deploying models in the real world. To simulate such challenges, several out-of-distribution benchmarks have been established, including measuring model performance

Table 1: Comparison of various baseline models. We replace the ResNet Bottleneck block with four different stronger building blocks, and keep the computational cost of baseline models on par with DeiT counterparts. We observe a significant gain in terms of clean image accuracy, but our CNN baseline models are still less robust than DeiT.

| Architecture | FLOPs | IN ($\uparrow$) | S-IN ($\uparrow$) | IN-C ($\downarrow$) | IN-R ($\uparrow$) | IN-SK ($\uparrow$) |
|---|---|---|---|---|---|---|
| ResNet50 | 4.1G | 78.4 | 9.9 | 51.2 | 39.2 | 27.7 |
| DeiT-S | 4.6G | 79.8 | 16.2 | 42.8 | 41.9 | 29.1 |
| ResNet-DW | 4.8G | 79.7 | 10.6 | 50.1 | 41.3 | 29.1 |
| ResNet-Inverted-DW | 4.6G | 80.0 | 10.8 | 47.7 | 41.9 | 29.4 |
| ResNet-Up-Inverted-DW | 4.7G | 79.7 | 12.9 | 47.9 | 42.9 | 30.8 |
| ResNet-Down-Inverted-DW | 4.5G | 79.6 | 12.1 | 49.0 | 42.5 | 29.5 |

on images with common corruptions (Hendrycks & Dietterich, 2018) or with various renditions (Hendrycks et al., 2021). Recently, a set of works (Bai et al., 2021; Zhang et al., 2022; Paul & Chen, 2022) find that Transformers are inherently much more robust than CNNs on out-of-distribution samples. Our work is a direct follow-up of Bai et al. (2021), whereas we stand on the opposite side—*we show CNNs can in turn outperform Transformers in out-of-distribution robustness*.

## 3 SETTINGS

Following the setup in Bai et al. (2021), in this paper, the robustness is thoroughly compared between CNNs and Transformers using ResNet (He et al., 2016) and ViT (Dosovitskiy et al., 2020).

**CNN Block Instantiations.** We consider four different block instantiations, as shown in Fig. 2. The first block is Depth-wise ResNet Bottleneck block (Xie et al., 2017), in which the 3×3 vanilla convolution layer is replaced with a 3×3 depth-wise convolution layer. The second block is the Inverted Depth-wise ResNet Bottleneck block, in which the hidden dimension is four times the input dimension (Sandler et al., 2018). The third block is based on the second block, with the depthwise convolution layer moved up in position as in ConvNeXT (Liu et al., 2022). Similarly, based on the second block, the fourth block moves down the position of the depthwise convolution layer. We replace the bottleneck building block in the original ResNet architecture with these four blocks, and refer to the resulting models as ResNet-DW, ResNet-Inverted-DW, ResNet-Up-Inverted-DW, and ResNet-Down-Inverted-DW, respectively.

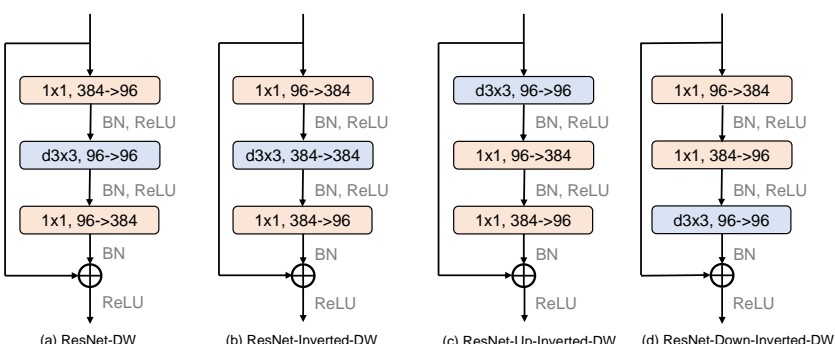

Figure 2: Architectures of four different instantiations of CNN block.

**Computational Cost.** In this work, we use FLOPs to measure model size. Here we note that directly replacing the ResNet bottleneck block with the above four instantiated blocks will significantly reduce the total FLOPs of the model, due to the usage of the depth-wise convolution. To mitigate the computational cost loss and to increase model performance, following the spirit of ResNeXT (Xie et al., 2017), we change the channel setting of each stage from (64, 128, 256, 512) to (96, 192, 384, 768). We then tune the block number in stage 3 to keep the total FLOPs of baseline models roughly the same as DeiT-S. The final FLOPs of our baseline models are shown in Tab. 1. Unless otherwise stated, all models considered in this work are of a similar scale as DeiT-S.

**Robustness Benchmarks.** In this work, we extensively evaluate the model performance on out-of-distribution robustness using the following benchmarks: 1) Stylized-ImageNet(Geirhos et al., 2018), which contains synthesized images with shape-texture conflicting cues; 2) ImageNet-C(Hendrycks & Dietterich, 2018), with various common image corruptions; 3) ImageNet-R (Hendrycks et al., 2021), which contains natural renditions of ImageNet object classes with different textures and local image statistics; 4) ImageNet-Sketch (Wang et al., 2019), which includes sketch images of the same ImageNet classes collected online.

**Training Recipe.** CNNs may achieve better robustness by simply tuning the training recipe (Bai et al., 2021; Wightman et al., 2021). Therefore, in this work, we deliberately apply the standard 300-epoch DeiT training recipe (Touvron et al., 2021a) for all models unless otherwise mentioned, so that the performance difference between models can only be attributed to the architecture difference.

**Baseline Results.** The results are shown in Tab. 1. For simplicity, we use "IN", "S-IN", "IN-C", "IN-R", and "IN-SK" as abbreviations for "ImageNet", "Stylized-ImageNet", "ImageNet-C", "ImageNet-R", and "ImageNet-Sketch". Same as the conclusion from (Bai et al., 2021), we can observe that DeiT-S shows much stronger robustness generalization than ResNet50. Moreover, we note that, even by equipping with stronger depth-wise convolution layers, the ResNet architecture is only able to achieve comparable accuracy on clean images while remaining significantly less robust than its DeiT counterpart. This finding indicates that the key to the impressive robustness of Vision Transformer lies in its architecture design.

## 4 COMPONENT DIAGNOSIS

In this section, we present three effective architectural designs that draw inspiration from Transformers, but work surprisingly well on CNNs as well. These designs are as follows: 1) patchifying input images (Sec. 4.1), b) enlarging the kernel size (Sec. 4.2), and finally, 3) reducing the number of activation layers and normalization layers (Sec. 4.3).

### 4.1 PATCHIFY STEM

A CNN or a Transformer typically down-samples the input image at the beginning of the network to obtain a proper feature map size. A standard ResNet architecture achieves this by using a 7×7 convolution layer with stride 2, followed by a 3×3 max pooling with stride 2, resulting in a 4× resolution reduction. On the other hand, ViT adopts a much more aggressive down-sampling strategy by partitioning the input image into p×p non-overlapping patches and projects each patch with a linear layer. In practice, this is implemented by a convolution layer with kernel size $p$ and stride $p$, where $p$ is typically set to 16. Here the layers preceding the typical ResNet block or the self-attention block in ViT are referred to as the *stem*. While previous works have investigated the importance of stem setup in CNNs (Trockman & Kolter, 2022) and Transformers (Xiao et al., 2021), none have examined this module from the perspective of robustness.

To investigate this further, we replace the ResNet-style stem in the baseline models with the ViT-style patchify stem. Specifically, we use a convolution layer with kernel size $p$ and stride $p$, where $p$ varies from 4 to 16. We keep the total stride of the model fixed, so that a 224×224 input image will always lead to a 7×7 feature map before the final global pooling layer. In particular, the original ResNet sets stride=2 for the first block in stages 2, 3, and 4. We set stride=1 for the first block in stage 2 when employing the 8×8 patchify stem, and set stride=1 for the first block in stages 2 and 3 when employing the 16×16 patchify stem. To ensure a fair comparison, we add extra blocks in stage 3 to maintain similar FLOPs as before.

In Tab. 2, we observe that increasing the patch size of the ViT-style patchify stem leads to increased performance on robustness benchmarks, albeit potentially at the cost of clean accuracy. Specifically, for all baseline models, when the patch size is set to 8, the performance on all robustness benchmarks is boosted by at least 0.6%. When the patch size is increased to 16, the performance on all robustness benchmarks is boosted by at least 1.2%, with the most significant improvement being 6.6% on Stylized-ImageNet. With these results, we can conclude that this simple patchify operation largely contributes to the strong robustness of ViT as observed in (Bai et al., 2021), and meanwhile, can play a vital role in closing the robustness gap between CNNs and Transformers.

Table 2: The performance of different baseline models equipped with ViT-style patchify stem. "PX" refers to the model having a ViT-style patchify stem with the patch size set to X. The block number in stage 3 is increased when the patch size is larger. With a larger patch size, the models attain better robustness but worse clean accuracy.

| Architecture | FLOPs | IN ($\uparrow$) | S-IN ($\uparrow$) | IN-C ($\downarrow$) | IN-R ($\uparrow$) | IN-SK ($\uparrow$) |
|---|---|---|---|---|---|---|
| ResNet50 | 4.1G | 78.4 | 9.9 | 51.2 | 39.2 | 27.7 |
| DeiT-S | 4.6G | 79.8 | 16.2 | 42.8 | 41.9 | 29.1 |
| ResNet-DW | 4.8G | 79.7 | 10.6 | 50.1 | 41.3 | 29.1 |
| + P4 | 4.6G | 79.6 | 11.4 | 51.2 | 40.5 | 28.3 |
| + P8 | 4.6G | 79.9 | 12.7 | 47.6 | 42.5 | 30.1 |
| + P16 | 4.6G | 78.5 | 17.0 | 44.3 | 44.8 | 32.0 |
| ResNet-Inverted-DW | 4.6G | 80.0 | 10.8 | 47.7 | 41.9 | 29.4 |
| + P4 | 4.5G | 80.2 | 11.2 | 48.8 | 41.5 | 29.8 |
| + P8 | 4.6G | 79.8 | 14.8 | 45.4 | 44.0 | 31.8 |
| + P16 | 4.6G | 77.9 | 17.2 | 42.5 | 44.9 | 32.4 |
| ResNet-Up-Inverted-DW | 4.7G | 79.7 | 12.9 | 47.9 | 42.9 | 30.8 |
| + P4 | 4.5G | 80.0 | 13.5 | 48.9 | 42.3 | 30.7 |
| + P8 | 4.7G | 79.8 | 15.7 | 46.3 | 44.0 | 31.4 |
| + P16 | 4.5G | 77.9 | 17.0 | 43.9 | 44.6 | 32.0 |
| ResNet-Down-Inverted-DW | 4.5G | 79.6 | 12.1 | 49.0 | 42.5 | 29.5 |
| + P4 | 4.4G | 79.8 | 12.4 | 48.6 | 41.8 | 29.4 |
| + P8 | 4.5G | 79.6 | 15.2 | 46.2 | 43.5 | 30.7 |
| + P16 | 4.6G | 78.0 | 16.2 | 43.8 | 43.8 | 30.6 |

We also conduct experiments with more advanced patchify stems. Surprisingly, while these stems improve the corresponding clean image accuracy, we find that they contribute little to out-of-distribution robustness. Further details can be found in Appendix A.2. This observation suggests that clean accuracy and out-of-distribution robustness do not always exhibit a positive correlation. In other words, designs that enhance clean accuracy may not necessarily result in better robustness. highlighting the importance of exploring methods that can improve robustness in addition to enhancing clean accuracy.

## 4.2 LARGE KERNEL SIZE

One critical property that distinguishes the self-attention operation from the classic convolution operation is its ability to operate on the entire input image or feature map, resulting in a global receptive field. The importance of capturing long-range dependencies has been demonstrated for CNNs even before the advent of Vision Transformer. A notable example is Non-local Neural Network, which has been shown to be highly effective for both static and sequential image recognition, even when equipped with only one non-local block (Wang et al., 2018). However, the most commonly used approach in CNN is still to stack multiple $3\times3$ convolution layers to gradually increase the receptive field of the network as it goes deeper.

In this section, we aim to mimic the behavior of the self-attention block, by increasing the kernel size of the depth-wise convolution layer. We experiment with various kernel sizes, including 5, 7, 9, 11, and 13, and evaluate their performance on different robustness benchmarks, are shown in Fig. 3. Our findings suggest that larger kernel sizes generally lead to better clean accuracy and stronger robustness. Nevertheless, we also observe that the performance gain gradually saturates when the kernel size becomes too large.

It is noteworthy that using (standard) convolutions with larger kernels will result in a significant increase in computation. For example, if we directly change the kernel size in ResNet50 from 3 to 5, the total FLOPs of the resulting model would be 7.4G, which is considerably larger than its Transformer counterpart. However, with the usage of depth-wise convolution layer, increasing the kernel size from 3 to 13 typically only increases the FLOPs by 0.3G, which is relatively small compared to the FLOPs of DeiT-S (4.6G). The only exceptional case here is ResNet-Inverted-DW: due to the large channel dimension in its Inverted Bottleneck design, increasing the kernel size from 3 to 13 results in a total increase of 1.4G FLOPs, making it an unfair comparison to some extent.

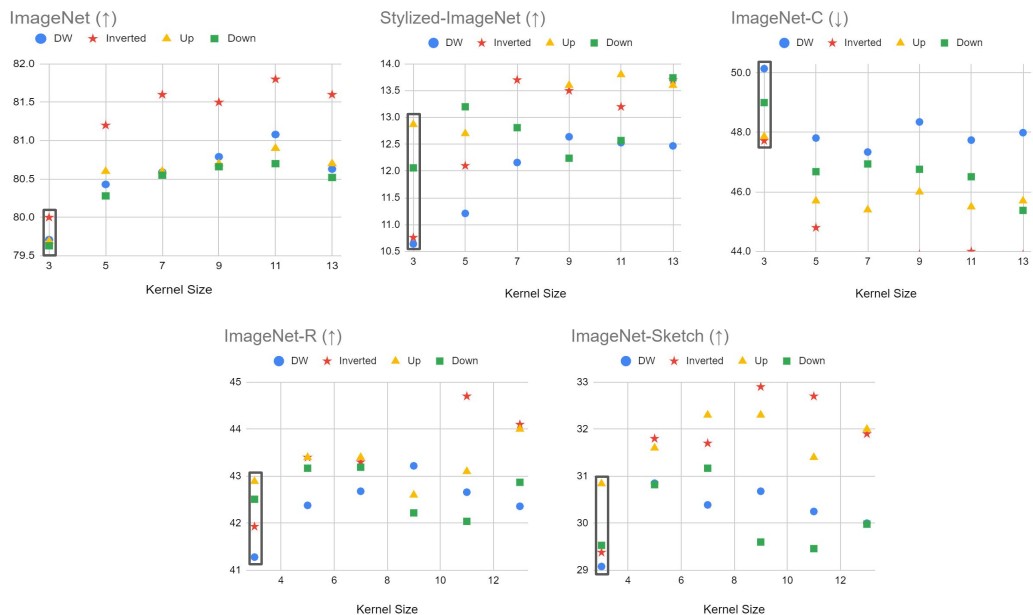

Figure 3: Robustness evaluation of model with varying kernel size. Different colors and shapes denote different models.

Table 3: The performance of models with differently removed activation layers. We use ✓ to denote keeping an activation layer and ✗ to denote removing an activation layer.

| Architecture | FLOPs | ReLU | | | IN ↑ | S-IN ↑ | IN-C ↓ | IN-R ↑ | IN-SK ↑ |
|---|---|---|---|---|---|---|---|---|---|
| | | $1^{st}$ | $2^{nd}$ | $3^{rd}$ | | | | | |
| ResNet50 | 4.1G | ✓ | ✓ | ✓ | 78.4 | 9.9 | 51.2 | 39.2 | 27.7 |
| DeiT-S | 4.6G | ✓ | ✓ | ✓ | 79.8 | 16.2 | 42.8 | 41.9 | 29.1 |
| ResNet-DW | 4.8G | ✓ | ✓ | ✓ | 79.7 | 10.6 | 50.1 | 41.3 | 29.1 |
| | | ✓ | ✗ | ✗ | 74.8 | 8.3 | 58.6 | 37.1 | 24.5 |
| | | ✗ | ✓ | ✗ | 73.2 | 8.6 | 60.5 | 36.2 | 23.3 |
| | | ✗ | ✗ | ✓ | 79.4 | 11.7 | 50.5 | 42.8 | 29.8 |
| ResNet-Inverted-DW | 4.6G | ✓ | ✓ | ✓ | 80.0 | 10.8 | 47.7 | 41.9 | 29.4 |
| | | ✓ | ✗ | ✗ | 80.7 | 12.3 | 46.4 | 45.3 | 31.9 |
| | | ✗ | ✓ | ✗ | 80.3 | 11.2 | 48.5 | 44.0 | 30.9 |
| | | ✗ | ✗ | ✓ | 72.5 | 9.7 | 59.5 | 37.5 | 24.7 |
| ResNet-Up-Inverted-DW | 4.7G | ✓ | ✓ | ✓ | 79.7 | 12.9 | 47.9 | 42.9 | 30.8 |
| | | ✓ | ✗ | ✗ | 67.8 | 9.4 | 65.0 | 33.0 | 19.7 |
| | | ✗ | ✓ | ✗ | 80.1 | 13.0 | 47.3 | 44.2 | 30.0 |
| | | ✗ | ✗ | ✓ | 68.7 | 8.9 | 64.4 | 33.4 | 20.2 |
| ResNet-Down-Inverted-DW | 4.5G | ✓ | ✓ | ✓ | 79.6 | 12.1 | 49.0 | 42.5 | 29.5 |
| | | ✓ | ✗ | ✗ | 80.7 | 13.0 | 46.6 | 45.9 | 32.8 |
| | | ✗ | ✓ | ✗ | 70.1 | 9.9 | 63.3 | 35.2 | 21.8 |
| | | ✗ | ✗ | ✓ | 68.8 | 8.3 | 65.1 | 33.7 | 21.9 |

As a side note, the extra computational cost incurred by a large kernel size can be mitigated by employing a patchify stem with a large patch size. As a result, our final models will still be on the same scale as DeiT-S. For models with multiple proposed designs, please refer to Sec. 5.

### 4.3 REDUCING ACTIVATION AND NORMALIZATION LAYERS

In comparison to a ResNet block, a typical Vision Transformer block has fewer activation and normalization layers (Dosovitskiy et al., 2020; Liu et al., 2021). This architecture design choice has also been found to be effective in improving the performance of ConvNeXT (Liu et al., 2022). Inspired by this, we adopt the idea of reducing the number of activation and normalization layers in

Table 4: The performance of models with differently removed activation and normalization layer. Note that the results are obtained with the best choice of only one activation layer in a block based on the results of Tab. 3. We use ✓ to denote keeping a normalization layer and ✗ to denote removing a normalization layer. It is observed that the highlighted architectural design choices consistently lead to improved robustness upon the baselines.

| Architecture | FLOPs | BN | | | IN ↑ | S-IN ↑ | IN-C ↓ | IN-R ↑ | IN-SK ↑ |
| --- | --- | --- | --- | --- | --- | --- | --- | --- | --- |
| | | $1^{st}$ | $2^{nd}$ | $3^{rd}$ | | | | | |
| ResNet50 | 4.1G | ✓ | ✓ | ✓ | 78.4 | 9.9 | 51.2 | 39.2 | 27.7 |
| DeiT-S | 4.6G | ✓ | ✓ | ✓ | 79.8 | 16.2 | 42.8 | 41.9 | 29.1 |
| ResNet-DW | 4.8G | ✓ | ✓ | ✓ | 79.7 | 10.6 | 50.1 | 41.3 | 29.1 |
| | | ✓ | ✗ | ✗ | 79.4 | 11.6 | 49.3 | 43.3 | 29.9 |
| | | ✗ | ✓ | ✗ | 79.5 | 10.6 | 49.5 | 43.1 | 29.3 |
| | | ✗ | ✗ | ✓ | 79.3 | 11.1 | 50.7 | 42.5 | 30.2 |
| ResNet-Inverted-DW | 4.6G | ✓ | ✓ | ✓ | 80.0 | 10.8 | 47.7 | 41.9 | 29.4 |
| | | ✓ | ✗ | ✗ | 80.4 | 12.9 | 47.0 | 45.8 | 32.4 |
| | | ✗ | ✓ | ✗ | 80.0 | 11.2 | 49.3 | 44.1 | 31.2 |
| | | ✗ | ✗ | ✓ | 80.1 | 13.0 | 46.8 | 44.3 | 30.8 |
| ResNet-Up-Inverted-DW | 4.7G | ✓ | ✓ | ✓ | 79.7 | 12.9 | 47.9 | 42.9 | 30.8 |
| | | ✓ | ✗ | ✗ | 80.5 | 14.3 | 45.0 | 47.2 | 33.1 |
| | | ✗ | ✓ | ✗ | 80.4 | 12.9 | 47.0 | 45.8 | 32.4 |
| | | ✗ | ✗ | ✓ | 80.1 | 13.0 | 46.8 | 44.3 | 30.8 |
| ResNet-Down-Inverted-DW | 4.5G | ✓ | ✓ | ✓ | 79.6 | 12.1 | 49.0 | 42.5 | 29.5 |
| | | ✓ | ✗ | ✗ | 80.5 | 13.6 | 46.4 | 45.9 | 33.3 |
| | | ✗ | ✓ | ✗ | 80.7 | 13.1 | 46.1 | 45.5 | 31.7 |
| | | ✗ | ✗ | ✓ | 80.8 | 12.7 | 46.5 | 46.1 | 33.1 |

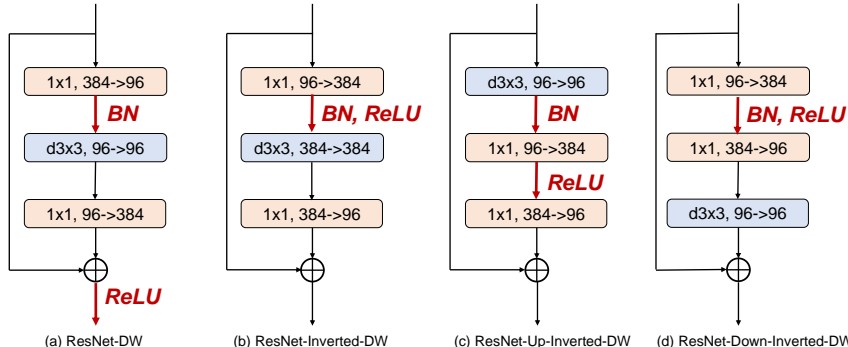

(a) ResNet-DW  (b) ResNet-Inverted-DW  (c) ResNet-Up-Inverted-DW  (d) ResNet-Down-Inverted-DW

Figure 4: The optimal position of normalization and activation layers for different block architectures.

all our four block instantiations for exploring its effects on robustness generalization. Specifically, a ResNet block typically contains one normalization layer and one activation layer following each convolution layer, resulting in a total of three normalization and activation layers in one block. In our implementation, we remove two normalization layers and activation layers from each block, resulting in only one normalization layer and activation layer.

We conduct experiments with different combinations of removing activation and normalization layers and empirically find that the best results are achieved by leaving only one activation layer after the convolution layer where the channel dimension expands (*i.e.*, the number of output channels is larger than that of input channels) and leaving one normalization layer after the first convolution layer. The optimal configuration is visualized in Fig. 4. The results of the models with differently removed layers are shown in Tab. 3 and Tab. 4. For example, for ResNet-Up-Inverted-DW, we observe considerable improvements of 1.4% on Stylized-ImageNet (14.3% *vs.* 12.9%), 2.9% on ImageNet-C (45.0% *vs.* 47.9%), 4.3% on ImageNet-R (47.2% *vs.* 42.9%), and 2.3% on ImageNet-Sketch (33.1% *vs.* 30.8%). Moreover, reducing the number of normalization layers results in lower GPU memory usage and faster training (Gitman & Ginsburg, 2017; Brock et al., 2021), with up to a 23% speed-up achieved by simply removing a few normalization layers.

Table 5: The performance of different baseline models equipped with ViT-style patchify stem, large kernel size, and fewer activation and normalization layers. The model with the suffix "PX" refers to that it has a ViT-style patchify stem with patch size set to X. "KX" refers to that the model uses blocks with a depth-wise convolution layer with kernel size X. "NormXActY" refers to that the model has only one normalization layer after the Xth convolution layer and one activation layer after the Yth convolution layer in the block.

| Architecture | FLOPs | IN (↑) | S-IN (↑) | IN-C (↓) | IN-R (↑) | IN-SK (↑) |
|---|---|---|---|---|---|---|
| ResNet50 | 4.1G | 78.4 | 9.9 | 51.2 | 39.2 | 27.7 |
| DeiT-S | 4.6G | 79.8 | 16.2 | 42.8 | 41.9 | 29.1 |
| ResNet-DW | 4.8G | 79.7 | 10.6 | 50.1 | 41.3 | 29.1 |
| + P16 + K11 + Norm1Act3 | 4.5G | 79.4 | 18.6 | 42.3 | 45.9 | 33.0 |
| ResNet-Inverted-DW | 4.6G | 80.0 | 10.8 | 47.7 | 41.9 | 29.4 |
| + P16 + K7 + Norm1Act1 | 4.6G | 79.0 | 19.5 | 42.1 | 45.9 | 32.8 |
| ResNet-Up-Inverted-DW | 4.7G | 79.7 | 12.9 | 47.9 | 42.9 | 30.8 |
| + P16 + K11 + Norm1Act2 | 4.4G | 79.2 | 20.2 | 40.9 | 48.7 | 35.2 |
| ResNet-Down-Inverted-DW | 4.5G | 79.6 | 12.1 | 49.0 | 42.5 | 29.5 |
| + P16 + K11 + Norm1Act1 | 4.6G | 79.9 | 19.3 | 41.6 | 46.0 | 32.8 |

Table 6: The robustness generalization of different models when DeiT-S serves as the teacher model. The results of DeiT-S here are included for reference; it is trained by the default recipe without knowledge distillation.

| Student Architecture | FLOPs | IN (↑) | S-IN (↑) | IN-C (↓) | IN-R (↑) | IN-SK (↑) |
|---|---|---|---|---|---|---|
| DeiT-S | 4.6G | 79.8 | 16.2 | 42.8 | 41.9 | 29.1 |
| ResNet-DW | 4.8G | 79.5 | 10.2 | 50.8 | 40.9 | 28.9 |
| **Robust**-ResNet-DW | 4.5G | 79.3 | 20.1 | 41.2 | 46.5 | 33.2 |
| ResNet-Inverted-DW | 4.6G | 80.1 | 11.0 | 47.9 | 42.4 | 30.4 |
| **Robust**-ResNet-Inverted-DW | 4.6G | 79.1 | 19.4 | 42.4 | 45.9 | 33.0 |
| ResNet-UpInvertedDW | 4.7G | 79.9 | 14.1 | 48.4 | 43.6 | 31.0 |
| **Robust**-ResNet-Up-Inverted-DW | 4.4G | 79.3 | 19.6 | 41.1 | 49.3 | 35.4 |
| ResNet-DownInvertedDW | 4.5G | 79.7 | 12.3 | 48.5 | 42.3 | 30.4 |
| **Robust**-ResNet-Down-Inverted-DW | 4.6G | 79.9 | 18.9 | 41.0 | 46.4 | 33.5 |

## 5 COMPONENTS COMBINATION

In this section, we explore the impact of combining all the proposed components on the model's performance. Specifically, we adopt a 16×16 patchify stem and an 11×11 kernel size, along with the corresponding optimal position for placing the normalization and activation layer for all architectures. An exception here is ResNet-Inverted-DW, for which we use 7×7 kernel size, as we empirically found that using a too-large kernel size (*e.g.*, 11×11) can lead to unstable training

As shown in Tab. 5, and Tab. 9, Tab. 10, Tab. 11 in Appendix A.3, *we can see these simple designs not only work well when individually applied to ResNet, but work even better when used together*. Moreover, by employing all three designs, ResNet now outperforms DeiT on all four out-of-distribution benchmarks. These results confirm the effectiveness of our proposed architecture designs and suggest that a pure CNN without any self-attention-like block can achieve robustness as good as ViT.

## 6 KNOWLEDGE DISTILLATION

Knowledge Distillation is a technique used for training a weaker student model with lower capacity by transferring the knowledge of a stronger teacher model. Typically, the student model can achieve similar or even better performance than the teacher model through knowledge distillation.

However, applying knowledge distillation directly to let ResNet-50 (the student model) learn from DeiT-S (the teacher model) has been found to be less effective in enhancing robustness as shown in (Bai et al., 2021). Surprisingly, when the model roles are switched, the student model DeiT-S remarkably outperforms the teacher model ResNet-50 on a range of robustness benchmarks, leading to the conclusion that the key to achieving good robustness of DeiT lies in its architecture, which therefore cannot be transferred to ResNet through knowledge distillation.

To investigate this further, we repeat these experiments with models that combine all three proposed architectural designs as the student models and with DeiT-S as the teacher model. As shown in Tab. 6, we observe that with the help of our proposed architectural elements from ViT, our resulting Robust-ResNet family now can consistently perform better than DeiT on out-of-distribution samples. In contrast, the baseline models, despite achieving good performance on clean ImageNet, are not as robust as the teacher model DeiT-S.

# 7 LARGER MODELS

To demonstrate the effectiveness of our proposed models on larger scales, we conduct experiments to match the total FLOPs of DeiT-Base. Specifically, we increase the base channel dimensions to (128, 256, 512, and 1024) and add over 20 blocks in stage 3 of the network, while using the ConvNeXT-B recipe for training (Liu et al., 2022) [1]. We compare the performance of the adjusted models to DeiT-B, as shown in Tab. 7. Our results indicate that our proposed Robust-ResNet family performs favorably against DeiT-B even on larger scales, suggesting that our method has great potential for scaling up models.

Table 7: Comparing the robustness of Robust-ResNet and DeiT at a larger scale.

| Architecture | FLOPs | IN ($\uparrow$) | S-IN ($\uparrow$) | IN-C ($\downarrow$) | IN-R ($\uparrow$) | IN-SK ($\uparrow$) |
|---|---|---|---|---|---|---|
| ResNet-200 | 15.1G | 81.7 | 13.8 | 42.2 | 45.1 | 34.5 |
| DeiT-B | 17.6G | 81.8 | 19.6 | 37.9 | 44.7 | 32.0 |
| **Robust**-ResNet-DW-B | 17.2G | 80.5 | 19.2 | 38.8 | 46.6 | 34.2 |
| **Robust**-ResNet-Inverted-DW-B | 17.1G | 81.6 | 22.1 | 35.5 | 50.2 | 36.7 |
| **Robust**-ResNet-Up-Inverted-DW-B | 17.1G | 81.9 | 23.1 | 35.1 | 50.5 | 37.1 |
| **Robust**-ResNet-Down-Inverted-DW-B | 17.3G | 81.3 | 22.2 | 35.6 | 49.7 | 37.8 |

# 8 CONCLUSION

The recent study by (Bai et al., 2021) claims that Transformers outperform CNNs on out-of-distribution samples, with the self-attention-like architectures being the main contributing factor. In contrast, this paper examines the Transformer architecture more closely and identifies several beneficial designs beyond the self-attention block. By incorporating these designs into ResNet, we have developed a CNN architecture that can match or even surpass the robustness of a Vision Transformer model of comparable size. We hope our findings prompt researchers to reevaluate the robustness comparison between Transformers and CNNs, and inspire further investigations into developing more resilient architecture designs.

## ACKNOWLEDGEMENT

This work is supported by a gift from Open Philanthropy, TPU Research Cloud (TRC) program, and Google Cloud Research Credits program.

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

# A APPENDIX

## A.1 TRAINING RECIPE DETAILS

We follow the standard 300-epoch DeiT training recipe (Touvron et al., 2021a) in this work. Specifically, we train all models using AdamW optimizer (Loshchilov & Hutter, 2019). We set the initial base learning rate to 5e-4, and apply the cosine learning rate scheduler to decrease it. Besides weight decay, we additionally adopt six data augmentation & regularization strategies (*i.e.*, RandAug (Cubuk et al., 2020), MixUp (Zhang et al., 2018), CutMix (Yun et al., 2019), Random Erasing (Zhong et al., 2020), Stochastic Depth (Huang et al., 2016), and Repeated Augmentation (Hoffer et al., 2020)) to regularize training.

## A.2 MORE STEM EXPERIMENTS

Recent studies suggest replacing the ViT-style patchify stem with a small number of stacked 2-stride 3×3 convolution layers can greatly ease optimization and thus boost the clean accuracy (Graham et al., 2021; Xiao et al., 2021). To verify its effectiveness on robustness benchmarks, we have also implemented the convolution stem for ViT-S in (Xiao et al., 2021), with a stack of 4 3×3 convolution layers with stride 2. The results are presented in Tab. 8. Surprisingly, though the use of convolutional stem does attain higher clean accuracy, it appears to be not as helpful as ViT-style patchify stem on out-of-distribution robustness.

Table 8: The performance of different baseline models equipped with ViT-style patchify stem and convolution stem. "PX" refers to that the model has a ViT-style patchify stem with patch size set to X; "C" refers to that it has a convolutional stem.

| Architecture | FLOPs | IN (↑) | S-IN (↑) | IN-C (↓) | IN-R (↑) | IN-SK (↑) |
|---|---|---|---|---|---|---|
| ResNet50 | 4.1G | 78.4 | 9.9 | 51.2 | 39.2 | 27.7 |
| DeiT-S | 4.6G | 79.8 | 16.2 | 42.8 | 41.9 | 29.1 |
| ResNet-DW | 4.8G | 79.7 | 10.6 | 50.1 | 41.3 | 29.1 |
| + P16 | 4.6G | 78.5 | 17.0 | 44.3 | 44.8 | 32.0 |
| + C | 4.5G | 79.6 | 16.2 | 48.4 | 43.0 | 30.2 |
| ResNet-Inverted-DW | 4.6G | 80.0 | 10.8 | 47.7 | 41.9 | 29.4 |
| + P16 | 4.6G | 78.1 | 17.4 | 42.9 | 44.7 | 32.4 |
| + C | 4.6G | 79.9 | 16.9 | 46.2 | 43.2 | 31.2 |
| ResNet-Up-Inverted-DW | 4.7G | 79.7 | 12.9 | 47.9 | 42.9 | 30.8 |
| + P16 | 4.5G | 77.9 | 17.0 | 43.9 | 44.6 | 32.0 |
| + C | 4.5G | 79.1 | 16.6 | 48.4 | 43.2 | 30.3 |
| ResNet-Down-Inverted-DW | 4.5G | 79.6 | 12.1 | 49.0 | 42.5 | 29.5 |
| + P16 | 4.6G | 78.0 | 16.2 | 43.8 | 43.8 | 30.6 |
| + C | 4.6G | 79.3 | 16.4 | 49.0 | 42.4 | 29.2 |

Table 9: The performance of different baseline models equipped with ViT-style patchify stem and large kernel size. The models with the suffix "PX" refers to that it has a ViT-style patchify stem with patch size set to X. "KX" refers to that the model uses blocks with a depth-wise convolution layer with kernel size X.

| Architecture | FLOPs | IN (↑) | S-IN (↑) | IN-C (↓) | IN-R (↑) | IN-SK (↑) |
|---|---|---|---|---|---|---|
| ResNet50 | 4.1G | 78.4 | 9.9 | 51.2 | 39.2 | 27.7 |
| DeiT-S | 4.6G | 79.8 | 16.2 | 42.8 | 41.9 | 29.1 |
| ResNet-DW | 4.8G | 79.7 | 10.6 | 50.1 | 41.3 | 29.1 |
| + P16 + K11 | 4.5G | 79.0 | 18.8 | 42.5 | 45.2 | 32.4 |
| ResNet-Inverted-DW | 4.6G | 80.0 | 10.8 | 47.7 | 41.9 | 29.4 |
| + P16 + K7 | 4.6G | 79.2 | 19.8 | 41.1 | 46.4 | 33.7 |
| ResNet-Up-Inverted-DW | 4.7G | 79.7 | 12.9 | 47.9 | 42.9 | 30.8 |
| + P16 + K11 | 4.4G | 78.0 | 17.2 | 43.7 | 47.1 | 31.2 |
| ResNet-Down-Inverted-DW | 4.5G | 79.6 | 12.1 | 49.0 | 42.5 | 29.5 |
| + P16 + K11 | 4.6G | 78.1 | 18.9 | 43.7 | 42.9 | 29.6 |

## A.3 MORE COMPONENTS COMBINATION EXPERIMENTS

Here we show more results of combining different components proposed in this paper in Tab. 9, Tab. 10, and Tab. 11.

## A.4 MORE COMPARISON WITH OTHER MODELS

Besides DeiT, here we also evaluate our proposed Robust-ResNet against two state-of-the-art models, ConvNeXt (Liu et al., 2022) and Swin-Transformer (Liu et al., 2021), in terms of out-of-distribution robustness. As shown in Tab. 12, all four of our models perform similarly to, or better than ConvNeXt or Swin-Transformer in all out-of-distribution tests.

## A.5 REPEATING EXPERIMENTS

To demonstrate the statistical significance of the robustness improvements achieved by our proposed components, we conduct three runs with different random seeds and report the mean and standard deviation in Tab. 13. We observe only small variations across the three runs, which confirms the consistent and reliable performance gains achieved by our proposed models.

Table 10: The performance of different baseline models equipped with ViT-style patchify stem and fewer activation and normalization layers. The model with the suffix "PX" refers to that it has a ViT-style patchify stem with patch size set to X. "NormXActY" refers to that the model has only one normalization layer after the Xth convolution layer and one activation layer after the Yth convolution layer in the block.

| Architecture | FLOPs | IN (↑) | S-IN (↑) | IN-C (↓) | IN-R (↑) | IN-SK (↑) |
|---|---|---|---|---|---|---|
| ResNet50 | 4.1G | 78.4 | 9.9 | 51.2 | 39.2 | 27.7 |
| DeiT-S | 4.6G | 79.8 | 16.2 | 42.8 | 41.9 | 29.1 |
| ResNet-DW | 4.8G | 79.7 | 10.6 | 50.1 | 41.3 | 29.1 |
| + P16 + Norm1Act3 | 4.6G | 78.7 | 16.2 | 43.0 | 46.3 | 33.1 |
| ResNet-Inverted-DW | 4.6G | 80.0 | 10.8 | 47.7 | 41.9 | 29.4 |
| + P16 + Norm1Act1 | 4.6G | 79.3 | 17.5 | 41.7 | 46.1 | 32.9 |
| ResNet-Up-Inverted-DW | 4.7G | 79.7 | 12.9 | 47.9 | 42.9 | 30.8 |
| + P16 + Norm1Act2 | 4.5G | 79.1 | 18.6 | 41.5 | 48.0 | 33.8 |
| ResNet-Down-Inverted-DW | 4.5G | 79.6 | 12.1 | 49.0 | 42.5 | 29.5 |
| + P16 + Norm1Act1 | 4.6G | 79.7 | 18.0 | 40.7 | 47.1 | 33.4 |

Table 11: The performance of different baseline models equipped with large kernel size and fewer activation and normalization layers. "KX" refers to that the model uses blocks with a depth-wise convolution layer with kernel size X. "NormXActY" refers to the model has only one normalization layer after the Xth convolution layer and one activation layer after the Yth convolution layer in the block.

| Architecture | FLOPs | IN (↑) | S-IN (↑) | IN-C (↓) | IN-R (↑) | IN-SK (↑) |
|---|---|---|---|---|---|---|
| ResNet50 | 4.1G | 78.4 | 9.9 | 51.2 | 39.2 | 27.7 |
| DeiT-S | 4.6G | 79.8 | 16.2 | 42.8 | 41.9 | 29.1 |
| ResNet-DW | 4.8G | 79.7 | 10.6 | 50.1 | 41.3 | 29.1 |
| + K11 + Norm1Act3 | 5.0G | 80.4 | 13.3 | 45.2 | 43.9 | 30.8 |
| ResNet-Inverted-DW | 4.6G | 80.0 | 10.8 | 47.7 | 41.9 | 29.4 |
| + K7 + Norm1Act1 | 5.0G | 80.8 | 13.1 | 46.7 | 45.7 | 32.1 |
| ResNet-Up-Inverted-DW | 4.7G | 79.7 | 12.9 | 47.9 | 42.9 | 30.8 |
| + K11 + Norm1Act2 | 4.9G | 81.4 | 15.8 | 42.9 | 48.3 | 33.8 |
| ResNet-Down-Inverted-DW | 4.5G | 79.6 | 12.1 | 49.0 | 42.5 | 29.5 |
| + K11 + Norm1Act1 | 4.6G | 81.3 | 14.2 | 43.6 | 46.1 | 32.8 |

Table 12: Robustness comparison against two state-of-the-art models, ConvNeXT and Swin-Transformer.

| Architecture | FLOPs | IN (↑) | S-IN (↑) | IN-C (↓) | IN-R (↑) | IN-SK (↑) |
|---|---|---|---|---|---|---|
| ConvNeXt-T | 4.5G | 82.1 | 18.9 | 41.6 | 47.2 | 33.9 |
| Swin-T | 4.5G | 81.4 | 15.5 | 48.4 | 41.3 | 29.3 |
| **Robust**-ResNet-DW | 4.5G | 79.4 | 18.6 | 42.3 | 45.9 | 33.0 |
| **Robust**-ResNet-Inverted-DW | 4.6G | 79.0 | 19.5 | 42.1 | 45.9 | 32.8 |
| **Robust**-ResNet-Up-Inverted-DW | 4.4G | 79.2 | 20.2 | 40.9 | 48.7 | 35.2 |
| **Robust**-ResNet-Down-Inverted-DW | 4.6G | 79.9 | 19.3 | 41.6 | 46.0 | 32.8 |

## A.6 IMAGENET-A EVALUATION

The ImageNet-A dataset comprises a set of natural adversarial samples that have a considerable negative impact on the performance of machine learning models. In Tab. 14, we compare the performance of our Robust-ResNet models and DeiT on the ImageNet-A dataset. Notably, while the Robust-ResNet models do not perform as well as DeiT with an input resolution of 224, increasing the input resolution (*e.g.*, to 320) significantly narrows the gap between Robust-ResNet and DeiT

Table 13: The results of repeating experiments three times with different random seeds. It can be observed that all Robust-ResNet variants bring statistically stable robustness improvement.

| Architecture | FLOPs | IN (↑) | S-IN (↑) | IN-C (↓) | IN-R (↑) | IN-SK (↑) |
|---|---|---|---|---|---|---|
| ResNet50 | 4.1G | 78.4 | 9.9 | 51.2 | 39.2 | 27.7 |
| DeiT-S | 4.6G | 79.8 | 16.2 | 42.8 | 41.9 | 29.1 |
| ResNet-DW | 4.8G | 79.7 | 10.6 | 50.1 | 41.3 | 29.1 |
| **Robust**-ResNet-DW | 4.5G | 79.3±0.1 | 19.2±0.6 | 41.7±0.6 | 46.0±0.1 | 32.6±0.3 |
| ResNet-Inverted-DW | 4.6G | 80.0 | 10.8 | 47.7 | 41.9 | 29.4 |
| **Robust**-ResNet-Inverted-DW | 4.6G | 79.2±0.2 | 19.8±0.6 | 41.9±0.2 | 45.8±0.1 | 32.6±0.1 |
| ResNet-Up-Inverted-DW | 4.7G | 79.7 | 12.9 | 47.9 | 42.9 | 30.8 |
| **Robust**-ResNet-Up-Inverted-DW | 4.4G | 79.3±0.1 | 19.6±0.4 | 41.2±0.4 | 48.7±0.1 | 35.0±0.1 |
| ResNet-Down-Inverted-DW | 4.5G | 79.6 | 12.1 | 49.0 | 42.5 | 29.5 |
| **Robust**-ResNet-Down-Inverted-DW | 4.6G | 79.7±0.1 | 19.5±0.2 | 41.8±0.2 | 46.0±0.1 | 33.0±0.1 |

Table 14: Robustness comparison on ImageNet-A.

| Model | Resolution | |
|---|---|---|
| | 224 | 320 |
| ResNet50 | 7.0 | 15.5 |
| DeiT-S | 19.9 | 23.6 |
| **Robust**-ResNet-DW | 12.3 | 19.9 |
| **Robust**-ResNet-Inverted-DW | 11.8 | 22.3 |
| **Robust**-ResNet-Up-Inverted-DW | 12.5 | 19.0 |
| **Robust**-ResNet-Down-Inverted-DW | 14.0 | 21.2 |

on ImageNet-A. We conjecture this is because objects of interest in ImageNet-A tend to be smaller than in standard ImageNet.

## A.7 STRUCTURAL RE-PARAMETERIZATION

A line of recent works promotes the idea of a training-time multi-branch but inference-time plain model architecture via structural re-parameterization (Ding et al., 2021; 2022a;b). RepLKNet (Ding et al., 2022b), in particular, has shown that re-parameterization using small kernels can alleviate the optimization issue associated with large-kernel convolution layers, without incurring additional inference costs. Given that the Robust-ResNet models also use large kernels, here we experiment with the idea of structural re-parameterization and leverage a training-time multi-branch block architecture to further enhance model performance. The block architectures are detailed in Fig. 5. The results of two different model scales, shown in Tab. 15 and Tab. 16, demonstrate generally improved performance with this re-parameterization approach. One exception may be Robust-ResNet-Up-Inverted-DW, which occasionally exhibits slightly worse robustness with re-parameterization. Notably, with the re-parameterization technique, we are able to train the Robust-ResNet-Inverted-DW model using a kernel size of 11.

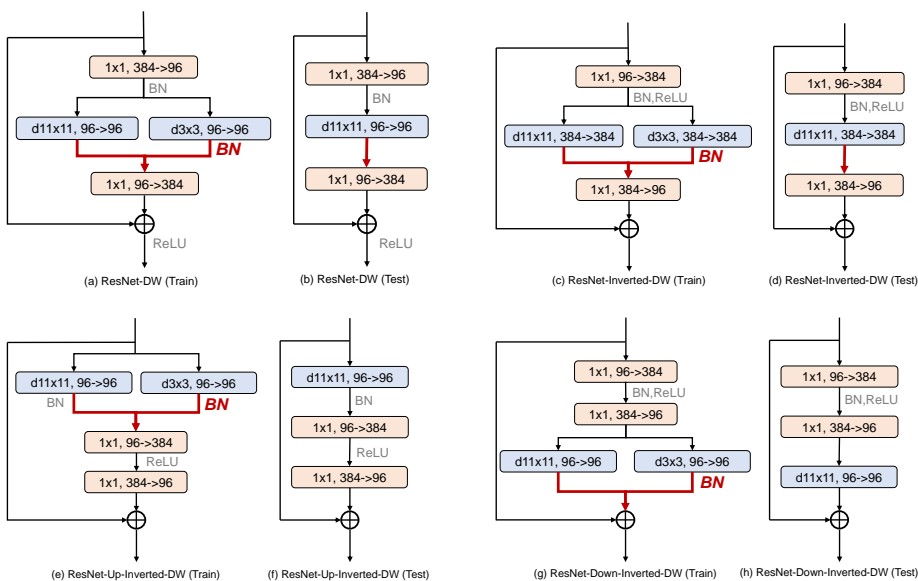

Figure 5: Architectures of four robust CNN block instantiations with structural reparameterization.

Table 15: The performance of Robust-ResNet models trained with structural re-parameterization technique at the scale of DeiT-S.

| Architecture | FLOPs | IN (↑) | S-IN (↑) | IN-C (↓) | IN-R (↑) | IN-SK (↑) |
|---|---|---|---|---|---|---|
| ResNet50 | 4.1G | 78.4 | 9.9 | 51.2 | 39.2 | 27.7 |
| DeiT-S | 4.6G | 79.8 | 16.2 | 42.8 | 41.9 | 29.1 |
| **Robust**-ResNet-DW | 4.5G | 79.4 | 18.6 | 42.3 | 45.9 | 33.0 |
| + Re-param | 4.5G | 79.6 | 19.0 | 41.1 | 47.8 | 34.2 |
| **Robust**-ResNet-Inverted-DW | 4.6G | 79.0 | 19.5 | 42.1 | 45.9 | 32.8 |
| + Re-param | 4.6G | 79.2 | 20.1 | 42.0 | 47.2 | 34.4 |
| **Robust**-ResNet-Up-Inverted-DW | 4.4G | 79.2 | 20.2 | 40.9 | 48.7 | 35.2 |
| + Re-param | 4.4G | 79.4 | 18.5 | 41.4 | 48.7 | 34.6 |
| **Robust**-ResNet-Down-Inverted-DW | 4.6G | 79.9 | 19.3 | 41.6 | 46.0 | 32.8 |
| + Re-param | 4.6G | 80.7 | 21.2 | 38.1 | 49.0 | 34.9 |

Table 16: The performance of Robust-ResNet models trained with structural re-parameterization technique at the scale of DeiT-B.

| Architecture | FLOPs | IN (↑) | S-IN (↑) | IN-C (↓) | IN-R (↑) | IN-SK (↑) |
|---|---|---|---|---|---|---|
| ResNet-200 | 15.1G | 81.7 | 13.8 | 42.2 | 45.1 | 34.5 |
| DeiT-B | 17.6G | 81.8 | 19.6 | 37.9 | 44.7 | 32.0 |
| **Robust**-ResNet-DW-B | 17.2G | 80.5 | 19.2 | 38.8 | 46.6 | 34.2 |
| + Re-param | 17.2G | 80.8 | 20.1 | 38.6 | 47.5 | 35.4 |
| **Robust**-ResNet-Inverted-DW-B | 17.1G | 81.6 | 22.1 | 35.5 | 50.2 | 36.7 |
| + Re-param | 17.1G | 82.1 | 23.1 | 35.4 | 50.7 | 37.4 |
| **Robust**-ResNet-Up-Inverted-DW-B | 17.1G | 81.9 | 23.1 | 35.1 | 50.5 | 37.1 |
| + Re-param | 17.1G | 82.2 | 22.9 | 34.5 | 50.6 | 37.8 |
| **Robust**-ResNet-Down-Inverted-DW-B | 17.3G | 81.3 | 22.2 | 35.6 | 49.7 | 37.8 |
| + Re-param | 17.3G | 81.6 | 22.0 | 34.9 | 51.1 | 38.1 |

