# OpenReview forum: "Can CNNs Be More Robust Than Transformers?"
_ICLR.cc/2023/Conference — ICLR 2023 poster_

### Official Review · Reviewer_DpNa · 2022-10-24

**Confidence:** 4
**Correctness:** 4
**Technical Novelty And Significance:** 3
**Empirical Novelty And Significance:** 3
**Recommendation:** 8

**Clarity, Quality, Novelty And Reproducibility:**

* To the best of my knowledge the experiments and the breaking down of ViT components is original.
* The writing, figures, tables and experiments were all of high quality and very clear.


**Strength And Weaknesses:**

**Strengths**
1. Well written with an easy to follow flow
2. I like the overall thought behind the paper of trying to figure out which components in ViTs have what amount of importance in getting better robustness. This type of paper shines some light on making better architectural decisions and understanding the effects of each component. Well done!
3. Experiments are well organized and easy to extract information from.

**Weaknesses**
1. Table 7 seems incomplete without results from unmodified ResNet.
2. The paper shows what matters through empirical evidence (and the experiments are interesting on their own), but it doesn't seem to provide any explanation/potential hypothesis about why patchifying/large kernel size/reducing #activation/normalization layers could improve robustness.

**Questions**
1. Table 4 description: What is meant by "It is observed that the optimal choice always lead to best robustness." I don't see that pattern.
2. All tables: How did you decide which row to highlight?
3. Is patchifying stem (i.e. non-overlapping convolution) only done in the first convolution operation of Robust-ResNet?
4. page 6 last paragraph: why such a drastic speed up from removing activation/normaliaztion layers?

**Typos**
1. page 2, paragraph 1: due to less normalization layers are used -> due to less normalization layers being used
2. page 3, paragraph 1: Transformers are inherently much more robustness than CNNs -> Transformers are inherently much more robust than CNNs
3. page 3, paragraph 1: last sentence was hard to read ("Our work is a direct follow-up...")
4. page 3, "Settings" section,  paragraph 2: in whichthe hidden -> in which the hidden
5. page 3, last paragraph: FLOating Point operations per second -> FLoating-point Operations Per Second
6. page 6, last paragraph: speep up training -> speed up training
7. page 7, table 3 description: block is keepped -> block is kept
8. page 7, table 4: always lead to -> always leads to
9. page 8, "Components Combination" section:  16×16 patchify stem with patch size ->  16×16 patchify stem with patch size 16
10. page 9, section 7: still performs favorably -> still perform favorably

**Summary Of The Paper:**

* **Motivation**: This paper takes a deep dive into why Vision Transformer (ViT) networks have empirically been found to be more robust than Convolutional Neural Networks (as evaluated by performance on out-of-distribution samples).

* **Approach**: They isolate some of the key components in DeiT that is missing in CNNs (besides the obvious attention mechanism), and do A/B experiments to figure out which components contribute the most to the differences in robustness. The components they evaluate are:
1. Patchifying Stem: DeiT partitions images into PxP non-overlapping patches. The authors replicate this in CNNs by applying non-overlapping convolutions.
2. Large Kernel Size: DeiT has better global receptive field. The authors replicate this in CNNs by increasing kernel size.
3. Reducing Activation and Normalization Layers: DeiT has fewer activation and normalization layers than a typical ResNet block. The authors replicate this by reducing #activation/normalization layers in the ResNet block.
The authors call the model with above 3 modifications a **Robust-ResNet**.

* **Findings**:
1. Patchifying has the most noticeable impact in improving robustness
2. Robust-ResNet is very similar in terms of #FLOPs and clean accuracy to both a DeiT and the (unmodified) ResNet. However, whereas the (unmodified) ResNet had noticeably worse robustness performance compared to DeiT, Robust-ResNet is able to achieve robustness results that are even slightly better than DeiT.
3. Whereas ResNet students (in model distillation) struggle in terms of robustness when the teacher is a ViT, Robust-ResNets seem to outperform the ViT teacher in terms of robustness.
4. Improved robustness seems to typically come at a cost to (clean) accuracy - E.g. Table 2

**Summary Of The Review:**

I recommend for this paper to be accepted because:
1. The strengths outlined above outweigh the weaknesses
2. I think the findings & experiments would be interesting to the community
3. This type of paper (A/B comparison of different components) helps in shining some light on the underlying workings of neural networks so that we are not just random searching for better architectures.

---

> ### Author Response · Authors · 2022-11-19
> **Response to Reviewer DpNa**
>
> We first thank the reviewer for the detailed comments and the appreciation of our work. We address the concerns below:
>
> **Q1: Results of unmodified ResNet**
>
> Thanks for your suggestion! Following Table 10 in [1], we will add the vanilla ResNet-200 as the baseline for comparison.
>
> **Q2: explanation/potential hypothesis about why these architectural elements work.**
>
> We agree that missing such explanations is a limitation of this paper. Nonetheless, we would like to highlight that our work is the first to comprehensively study the relationship between neural architectures and ood robustness at the micro level. We believe our results are interesting, worth being known to the research community, and important for motivating future empirical/theoretical works in this direction.
>
> **Q3: Tab.4 Description**
>
> Sorry for the misleading description. We will rephrase it to “It is observed that the highlighted choice always leads to improved robustness upon baseline”.
>
> **Q4: Table rows to highlight**
>
> Our tables highlight the design choices that are used in Robust-ResNet; we also use bold to mark the best performance.
>
> **Q5: Patchifying stem only in the first convolution operation?**
>
> Yes. We will make it clear in the next version.
>
> **Q6: Speed up from activation/normalization removal**
>
> The speedup mostly comes from removing normalization layers — as demonstrated in [2][3], removing normalization layers can decrease GPU memory consumption and accelerate network training. We will discuss these references accordingly in the next version.
>
> **Q7: Typos**
>
> Thanks for pointing out these typos! We will correct them and further polish the writing of this paper.
>
> [1] Bai, Yutong, et al. “Are Transformers More Robust Than CNNs.” NeurIPS 2021.
>
> [2] Brock, Andy, et al. "High-performance large-scale image recognition without normalization." ICML 2021.
>
> [3] Gitman, Igor, and Boris Ginsburg. "Comparison of batch normalization and weight normalization algorithms for the large-scale image classification." arXiv preprint 2017.

---

### Official Review · Reviewer_ntqh · 2022-10-24

**Confidence:** 4
**Correctness:** 3
**Technical Novelty And Significance:** 2
**Empirical Novelty And Significance:** 4
**Recommendation:** 8

**Clarity, Quality, Novelty And Reproducibility:**

Reasonably clear. The experiments should be reproducible based on the text. Novelty is not in techniques but more in the empirical analysis.

**Strength And Weaknesses:**

**Strengths**

- The paper conducts a thorough empirical evaluation in terms of different combination of the main design choices.
- It is interesting to look at the effect of architecture choices not just on accuracy but also robustness.

**Weaknesses**

- Most of the underlying design choices themselves have appeared in previous papers. As authors note, their work is very close to the ConvNeXT work, with the main difference being that their focus is on robustness.
- I missed an evaluation of ConvNeXT itself in the various tables. While the ConvNeXT paper did not focus on robustness, is it the case that the ConvNeXT models are also just as robust as the best models proposed in the paper? If so, then this paper's analysis would still be useful, but the narrative would need to be re-cast as understanding the robustness of ConvNeXT vs ViT.
- The paper lists accuracy numbers but without error-bars. It is difficult to gauge if the differences above are statistically significant. It would be worth repeating each experiment 3-5 times (at least for the -S models) and report both mean and standard deviation.
- The tables compare models in terms of FLOPs, but not in terms of number of parameters. Do the proposed models have a larger or smaller number of parameters? How does the accuracy compare when we compare to models with equivalent numbers of parameters? Again, it is reasonable to want models with equivalent FLOPs, but the #parameters analysis could reveal an alternate hypothesis for what makes the architectures robust (viz, number of parameters).
- The paper also misses an evaluation of adversarial robustness --- in terms of adversarial perturbations and to Imagenet-A (the so called natural adversarial examples). Given that the paper's contribution is largely empirical, these comparisons are needed for completeness.

**Summary Of The Paper:**

The paper seeks to understand architecture changes to traditional CNNs that render them as, or more, robust as Transformers when applied to out of distribution data. The main conclusions of the paper are that employing (a) an initial partition of the image into non-overlapping patches; (b) larger kernel sizes in a resnet block; and (c) having fewer non-linear layers (i.e., with normalizations and activations), can improve robustness.

**Summary Of The Review:**

Overall, this is an interesting empirical analysis of how architecture design choices can affect robustness, and specifically, of the elements of the Transformer architecture that can be 'back-ported' to CNNs to increase the latter's robustness.

However, since the main contribution is this empirical analysis, there is more of a burden for it to be complete, and as mentioned above, it is lacking in several respects.

### Post Rebuttal

The authors have provided a detailed response with several additional experiments that address all my concerns. I believe the paper should be accepted (I'd give it a 7 instead of 8, but since that's no longer an option, I believe it's closer to 8 than 6).

---

> ### Author Response · Authors · 2022-11-19
> **Response to reviewer ntqh (1/2)**
>
> We first thank the reviewer for the detailed comments. We address the concerns below:
>
> **Q1: novelty w.r.t. previous works**
>
> As stated in our general response, the main goal of this paper is to empirically understand the relationship between neural architecture and robustness at the micro level, i.e., finding which architectural elements are important for boosting model robustness. Though these architectural elements have already appeared in previous papers, but, to our best knowledge, no existing works revealed their “huge” potential in enhancing robustness.
>
> Our work is also substantially different from ConvNeXT. Rather than providing a strong but fixed neural architecture as in ConvNeXT, this paper aims to provide general suggestions to robustify (ideally any) CNNs. Moreover, rather than making 11 architectural modifications to arrive at the final design of ConvNeXT, our message is much simpler (with only 3 modifications) yet still powerful.
>
> Therefore we believe our findings are still novel and worth being known to the research community. We will make this part clear in the next version.
>
>
> **Q2: Comparison with ConvNeXT**
>
> Thanks for your suggestion. Below we compare the ood robustness of ConvNeXT-Tiny and our best model Robust-Resnet-Up-Invert-Small. We note our model consistently outperforms ConvNeXT in all ood tests by a clear margin. We will add and discuss these results accordingly in the next version.
>
> |             Model            | S-IN&uarr; | IN-C&darr; | IN-R&uarr; | IN-SK&uarr; |
> |:----------------------------:|:----:|:----:|:----:|:-----:|
> |           ConvNeXT-Tiny           | 18.9 | 41.6 | 47.2 |  33.9 |
> | Robust-ResNet-Up-Inverted-DW | 20.2 | 40.9 | 48.7 |  35.2 |
>
> **Q3: Repeat experiments**
>
> Thanks for such a golden suggestion! Here we report repeated Robust-Resnet results (originally in Table 5) with three different random seeds. We can observe that the variation is small, and the proposed models can consistently lead to stronger robustness. We will add this result in the next version.
>
>
> |              Model             |    IN&uarr;    |     S-IN&uarr;    |     IN-C&darr;    |     IN-R&uarr;    |    IN-SK&uarr;    |
> |:------------------------------:|:--------:|:--------:|:--------:|:--------:|:--------:|
> |            ResNet-DW           |   79.7   |   10.6   |   50.1   |   41.3   |   29.1   |
> |        Robust-ResNet-DW        | 79.3±0.1 | 19.2±0.6 | 41.7±0.6 | 46.0±0.1 | 32.6±0.3 |
> |       ResNet-Inverted-DW       |   80.0   |   10.8   |   47.7   |   41.9   |   29.4   |
> |    Robust-ResNet-Inverted-DW   | 79.2±0.2 | 19.8±0.6 | 41.9±0.2 | 45.8±0.1 | 32.6±0.1 |
> |      ResNet-Up-Inverted-DW     |   79.9   |   14.1   |   48.4   |   43.6   |   31.0   |
> |  Robust-ResNet-Up-Inverted-DW  | 79.3±0.1 | 19.6±0.4 | 41.2±0.4 | 48.7±0.1 | 35.0±0.1 |
> |     ResNet-Down-Inverted-DW    |   79.7   |   12.3   |   48.5   |   42.3   |   30.4   |
> | Robust-ResNet-Down-Inverted-DW | 79.7±0.1 | 19.5±0.2 | 41.8±0.2 | 46.0±0.1 | 33.0±0.1 |
>
> **Q4: Using the number of parameters to measure model scale**
>
> The core operation in ViT, self-attention, is a computation-intense but param-efficient operation. Therefore using the number of parameters as the metric to measure model capacity could lead to an unfair comparison. This is also the reason why prior works (e.g., [1][2]) mostly use FLOPs as the measure of model capacity.
>
> Nonetheless, as the reviewer suggested, we agree that comparing robustness under this situation is still interesting. We scale the number of parameters of DeiT-Small to the same level as our Robust-ResNet by increasing the embedded dimension, dubbed DeiT-Dim512. Note that this DeiT-Dim512 incurs considerably more computation cost than our Robust-ResNet models (~67% more FLOPs). As shown in the table below, compared to DeiT-Dim512, our Robust-ResNets can still attain on-par or stronger robustness on most ood tests.
>
> |              Model             | FLOPs (G) | Params(M) |  IN&uarr;  | S-IN&uarr; | IN-C&darr; | IN-R&uarr; | IN-SK&uarr; |
> |:------------------------------:|:---------:|:---------:|:----:|:----:|:----:|:----:|:-----:|
> |           DeiT-Dim512          |    7.5    |    38.9   | 80.8 | 17.2 | 40.4 | 42.8 |  30.3 |
> |        Robust-ResNet-DW        |    4.5    |    38.6   | 79.4 | 18.6 | 42.3 | 45.9 |  33.0 |
> |    Robust-ResNet-Inverted-DW   |    4.6    |    33.6   | 79.0 | 19.5 | 42.1 | 45.9 |  32.8 |
> |  Robust-ResNet-Up-Inverted-DW  |    4.4    |    34.4   | 79.2 | 20.2 | 40.9 | 48.7 |  35.2 |
> | Robust-ResNet-Down-Inverted-DW |    4.6    |    34.3   | 79.9 | 19.3 | 41.6 | 46.0 |  32.8 |

---

> > ### Author Response · Authors · 2022-11-19
> > **Response to reviewer ntqh (2/2)**
> >
> > **Q5: Adversarial robustness & ImageNet-A evaluation**
> >
> > We believe the adversarial robustness comparison between ResNet and DeiT is already well addressed in [3], i.e., the choice of activation function is the key for ensuring strong adversarial robustness. Therefore, we choose not to discuss such comparisons further in this paper.
> >
> > Regarding ImageNet-A, we found one trick to boost accuracy significantly: testing at a larger image resolution. We conjecture this is because objects of interest in ImageNet-A are smaller than in standard ImageNet. As shown in the table below, when testing at the image resolution = 320, our models are comparable to DeiT-S.
> >
> > We will add these discussions & results in the next version.
> >
> >
> > |        Model/Input Size        |  224  |  320  |
> > |:------------------------------:|:-----:|:-----:|
> > |             DeiT-S             | 19.9 | 23.6 |
> > |        Robust-ResNet-DW        | 12.3 | 19.9 |
> > |    Robust-ResNet-Inverted-DW   | 11.8 | 22.3 |
> > |  Robust-ResNet-Up-Inverted-DW  | 12.5 | 19.0 |
> > | Robust-ResNet-Down-Inverted-DW | 14.0 | 21.2 |
> >
> >
> > [1] Liu, Ze, et al. "Swin transformer: Hierarchical vision transformer using shifted windows." CVPR 2021.
> >
> > [2] Liu, Zhuang, et al. "A convnet for the 2020s." CVPR 2022.
> >
> > [3] Bai, Yutong, et al. “Are Transformers More Robust Than CNNs.” NeurIPS 2021.

---

### Official Review · Reviewer_rxfk · 2022-10-24

**Confidence:** 4
**Correctness:** 4
**Technical Novelty And Significance:** 3
**Empirical Novelty And Significance:** 3
**Recommendation:** 6

**Clarity, Quality, Novelty And Reproducibility:**

* The paper is well written and easy to follow
* Lack novelty in theory.
* Training details are provided in Appendix A. Should be easy to reproduce  most of the results in this work.

**Strength And Weaknesses:**

[Strength]

* The proposed method is able to improve the robustness of CNN models and make them comparable to DeiT-S in terms of robustness.
* The proposed method is validated on four popular ImageNet variants for robustness evaluation
* Experiments are reported in great detail, with sufficient ablation studies.

[Weakness]

* The paper is mostly empirical and lacks theoretical motivation or guarantee. The authors simply report the accuracy increment or decrement without further investigation. So it is a good-to-know fact but cannot inspire in-depth understanding of the robustness of CNN.

* The numerical experiments suggest that, the proposed three modifications, are not universal. Sometimes they can improve accuracies on ImageNet variants but sometimes dramatic performance drop can happen. What is worse, the patterns of good / bad cases are random. So the proposed method cannot transfer well. For a new given dataset, it is hard to tell whether we should apply the proposed method, or how we should choose the pattern in BN/ReLU removal.

* The proposed method will hurt the performance on clean datasets most of the time. This is a hard trade-off in practice.

* The robustness improvement over DeiT is not very significant for some networks and some datasets. Sometimes it is worse than DeiT (Table 1).

* Only DeiT is selected for comparison. Why DeiT? What about Swin-Transformer or other robust VIT models?


**Summary Of The Paper:**

The authors proposed three modifications for convolutional network to make it more robust: 1) split the input image into non-overlapping image patches; 2) use large kernel size; 3) remove some BN and ReLU layer. Numerical experiments on several ImageNet variant datasets validate the robustness improvement with slight degradation of the accuracy on clean datasets.


**Summary Of The Review:**

This work lacks novelty in theory. The proposed method is not universal and sometimes even degrade the performance dramatically.

---

> ### Author Response · Authors · 2022-11-19
> **To Reviewer rxfk**
>
> We first thank the reviewer for the detailed comments. We address the concerns below:
>
> **Q1: Mostly empirical & lacks theoretical motivation or guarantee.**
>
> As stated in our general response, the main goal of this paper is to empirically understand the relationship between neural architecture and robustness at the micro level, i.e., finding which architectural elements are important for boosting model robustness. We believe our results are interesting, worth being known to the research community, and important for motivating future empirical/theoretical works in this direction.
>
> **Q2: Not universal & random pattern**
>
> We respectfully disagree with it. As shown in our experiments, each modification can substantially improve robustness. Moreover, as shown in Tables 5 & 7, the final models attain much stronger robustness than DeiT when all proposed modifications are applied. Regarding ‘the pattern in BN/ReLU removal’, we observed that leaving one activation layer after the channel expanding layer and one normalization layer after the first convolution layer could be a general design principle. We will make it more clear in the next version.
>
> We hope this response could address your concerns. However, if not, please let us know which particular results (e.g., tables/figures) still concern you.
>
> **Q3: Clean performance drop & hard trade-off**
>
> Firstly, we would like to reiterate that the main focus of this paper is robustness on out-of-distribution data; as reported in our paper, our modifications successfully and significantly boost robustness. Though our individual modifications may lead to an accuracy drop on ImageNet, we would like to highlight that, as shown in Tables 5 & 7, our final models attain very similar ImageNet accuracy compared to baseline models (and more importantly, our robustness is much stronger).
>
> We hope this response could address your concerns. However, if not, please let us know which particular results (e.g., tables/figures) still concern you.
>
> **Q4: Even worse robustness than DeiT in Tab.1**
>
> We would like to point out that Table 1 is our vanilla ResNet baselines; it is expected that they have worse robustness than DeiT. Tables 5 & 7 are our Robust-ResNet models, which are comparable to or stronger than DeiT in robustness.
>
> **Q5: Comparison with Swin-Transformer**
>
> Our work is a direct follow-up of [1], which shows DeiT has much stronger robustness than ResNet. In this paper, we keep this model selection setup, while showing that, with proper modifications of ResNet, it can attain comparable or better robustness than DeiT.
>
> Per your request, we also add the results of Swin-Transformer below. We note that our Robust-Resnets still outperform Swin-Transformer on all ood datasets. We will add and discuss these results accordingly in the next version.
> |              Model             |  IN&uarr;  | S-IN&uarr; | IN-C&darr; | IN-R&uarr;  | IN-SK&uarr; |
> |:------------------------------:|:----:|:----:|:----:|:----:|:-----:|
> |            Swin-Tiny           | 81.4 | 15.5 | 48.4 | 41.3 |  29.3 |
> |        Robust-ResNet-DW        | 79.4 | 18.6 | 42.3 | 45.9 |  33.0 |
> |    Robust-ResNet-Inverted-DW   | 79.0 | 19.5 | 42.1 | 45.9 |  32.8 |
> |  Robust-ResNet-Up-Inverted-DW  | 79.2 | 20.2 | 40.9 | 48.7 |  35.2 |
> | Robust-ResNet-Down-Inverted-DW | 79.9 | 19.3 | 41.6 | 46.0 |  32.8 |

---

> > ### Comment · Reviewer_rxfk · 2022-11-21
> > **Thanks for the feedback**
> >
> > The feedback addressed most of my concerns. I updated my score.

---

> > > ### Author Response · Authors · 2022-11-22
> > > **Thanks for raising your score!**
> > >
> > > Thank you so much for raising your score. We are glad that your concerns get addressed!

---

### Author Response · Authors · 2022-11-19
**General Response**

We thank all the reviewers for their constructive comments! We are delighted to see that all reviewers agree that this paper conducts a thorough and well-organized empirical analysis of the out-of-distribution (ood) robustness of CNN vs. Transformer, with sufficient detail to be reproducible.

As suggested by the reviewers, we included the following experimental results in this rebuttal:

1.  Comparison with Swin-Transformer: we find that our Robust-ResNet consistently outperforms Swin-Transformer. See response to reviewer rxfk for details.

2. Comparison with ConvNeXt: our best model, Robust-ResNet-Up-Invert-DW, can consistently outperform ConvNeXt. See response to reviewer ntqh for details.

3. Statistics: we rerun all four Robust-ResNet with three different random seeds, and find the performance variation on ImageNet and ood is negligible, indicating a reliable robustness boost achieved by our proposed design choices. See response to reviewer ntqh for details.

4. The number of parameters as the metric: by scaling DeiT with a similar number of parameters as our Robust-ResNet, we note all four Robust-ResNet still attain comparable, or even better, ood robustness compared to this new DeiT. See response to reviewer ntqh for details.

5. Comparison on ImageNet-A: we found Robust-ResNet and DeiT attain similar performance if testing at the resolution=320. See response to reviewer ntqh for details.

Lastly, we would like to reiterate that the main goal of this paper is to empirically understand the relationship between neural architecture and robustness at the micro level, i.e., finding which architectural elements are important for boosting model robustness. We believe our results are interesting, worth being known to the research community, and important for motivating future empirical/theoretical works in this direction.

---

### Decision · Program_Chairs · 2023-01-20

**Decision:**

Accept: poster

**Justification For Why Not Higher Score:**

Mostly empirical work. Novelty low and theoretical understanding low.

**Justification For Why Not Lower Score:**

Reviewers still felt this paper carried enough useful information for publication.

**Metareview: Summary, Strengths And Weaknesses:**

Paper Summary:
This paper does a deep dive on architectural design considerations of CNNs, inspired by choices in transformers, and demonstrates that by changing CNN designs to be more similar with transformers, CNNs can outperform transformers without using self-attention mechanisms. The design choices involved include patchifying inputs, expanding receptive fields with depth-wise convolutions, and reducing the number of normalization and activation layers.

Review Summary:

Pros:
	- Well written (DpNa)
	- CNN performance improved relative to transformers such as DeIT (rxfk, DpNa)
	- Thorough evaluation (rxfk, ntqh,DpNa)
	- Robustness analysis is interesting (ntqh,DpNa)

Cons:
	- Mostly empirical work without sufficient understanding (rxfk,DpNa) -- authors confirm this but argue their empirical observations still have value to community.
	- Unclear if results are generalizable across datasets (rxfk)
	- Only DEiT selected for comparison. What about othr architectures? (rxfk) -- authors added Swin.
	- Many of these design choices have appeared in previous works (ntqh)
	- How does ConvNeXT compare with the robustness measures? (ntqh) -- Authors added experiments.
	- Statistical significance not reported (ntqh) -- variance now reported
	- What about number of parameters? (ntqh) -- added
	- What about adversarial robustness? (ntqh) -- authors commented that this is addressed in prior work



AC Recommendation: Accept. Reviewers strongly lean accept. Interesting experiments that are helpful generally when considering architecture design.

**Note From Pc:**

if the above contains the word "oral" or "spotlight" please see: "oral" presentation means -> notable-top-5% and "spotlight" means -> notable-top-25%. As stated in our emails, we are disassociating presentation type from AC recommendations